# Natural Allelic Variations of *Bch10G006400* Controlling Seed Size in Chieh-qua (*Benincasa hispida* Cogn. var. *Chieh-qua* How)

**DOI:** 10.3390/ijms25084236

**Published:** 2024-04-11

**Authors:** Yin Gao, Jiazhu Peng, Yanchun Qiao, Guoping Wang

**Affiliations:** 1Guangzhou Academy of Agricultural and Rural Sciences, Guangzhou 510335, China; yyingao@outlook.com; 2Guangzhou Academy of Agricultural Sciences, Guangzhou 510308, China; gajupeng@outlook.com; 3College of Horticulture, South China Agricultural University, Guangzhou 510642, China

**Keywords:** chieh-qua, seed size, bulk segregant analysis, fine-mapping, candidate gene

## Abstract

Seeds are the most important reproductive organs of higher plants, the beginning and end of a plant’s lifecycle. They are very important to plant growth and development, and also an important factor affecting yield. In this study, genetic analysis and BSA-seq of the F_2_ population crossed with the large-seeded material ‘J16’ and small-seeded material ‘FJ5’ were carried out, and the seed size locus was initially located within the 1.31 Mb region on chr10. In addition, 2281 F_2_ plants were used to further reduce the candidate interval to 48.8 Kb. This region contains only one gene encoding the N-acetyltransferase (NAT) protein (*Bch10G006400*). Transcriptome and expression analysis revealed that the gene was significantly more highly expressed in ‘J16’ than in ‘FJ5’. Variation analysis of *Bch10G006400* among parents and 50 chieh-qua germplasms revealed that as well as a nonsynonymous mutation (SNP_314) between parents, two mutations (SNP_400 and InDel_551) were detected in other materials. Combining these three mutations completely distinguished the seed size of the chieh-qua. GO and KEGG enrichment analyses revealed that DGEs played the most important roles in carbohydrate metabolism and plant hormone signal transduction, respectively. The results of this study provide important information for molecular marker-assisted breeding and help to reveal the molecular mechanism of seed size.

## 1. Introduction

Chieh-qua (*Benincasa hispida* Cogn. var. *chieh-qua* How), a variety of wax gourd, belongs to Cucurbit family. The fruit is rich in a variety of nutrients, usually grown in Southeast Asia and China. It has a combination of delicious and healthy qualities, and is popular with consumers [1].

Seed size is a very important trait. Large seeds can provide additional nutrients for the early growth of plants and are tolerant to abiotic stress [2]. Therefore, it is important to explore seed traits for improving crop germplasms, increasing yields, and solving bioenergy problems.

The development of seeds is controlled by the coordination of the outer skin, endosperm, and embryo. Seed size is influenced by genetic information from offspring and maternal genotypes [3,4], whose genetic features are very complicated. During double fertilization, progeny genotypes regulate the entire process [5], but the seed coat is controlled by the maternal genotype [6]. The size of a plant seed is regulated by a variety of signaling pathways, which have been identified in many crops [7], including the HAIKU (IKU) pathway, ubiquitin–proteasome pathway, G (Guanosine triphosphate) protein regulatory pathway, mitogen-activated protein kinase (MAPK) pathway, transcriptional regulators pathway, etc. In addition, many studies have shown that transcription factor and plant hormones also play important roles in seed size, e.g., the auxin, brassinosteroid (BR), gibberellin (GA), jasmonic acid (JA), cytokinin (CK), abscisic acid (ABA), and microRNA (miRNA) regulatory pathways. At present, the model plant *Arabidopsis thaliana* and the field crop rice have been deeply studied for seed size and 1000-seed weight, and the genes controlling seed size have been accurately located and cloned [8,9]. In addition, the research on seed size is also relatively in-depth for watermelon. Fine mapping of QTLs for seed-related traits in watermelon by CAPS, DCAP, and InDel markers was carried out, and major QTLs for seed-size related traits were detected within 55.5 Kb mapping interval, with a hypothesis that *CLA009291*, *CLA009301*, and *CLA009310* may be involved in the regulation of watermelon seed size [10]. Through the study of 197 watermelon seeds, the results showed that there was a high correlation between 100-seed weight of watermelon seeds and seed length, and two genes (*CIPP2C*, *CIARG1*) were identified, both of which have negative regulatory effects on plant ABA content [11,12,13,14]. Gene editing was used to knock out the β-glucosidase gene *CIBG1*, resulting in reduced ABA levels, a decrease in the number of cells within the seed, and a decrease in seed size and quality [15].

In other cucurbit crops, QTL mapping of seed length and width of Indian pumpkin was performed using 100 F_2_ populations obtained by crossing two parents of Indian pumpkin ‘2013-1’ and ‘9-6’. Four QTL loci controlling grain length were detected on four chromosomes (4, 6, 17, and 18), and the contribution rate was the highest at 38.6%. Four QTL loci controlling grain width were detected on four chromosomes (4, 5, 6, and 8), with a minimum contribution rate of 6.9% and a maximum contribution rate of 28.9% [16]. The seed size attributes of 75 pumpkin varieties were analyzed by GWAS, and four candidate genes related to seed size were identified (*Go-0044711*, *Go-0090407*, *Go-0004396*, and *Go-0019200*). These four genes increase the size of pumpkin seeds by regulating nutrient synthesis and macromolecular metabolism in plants [17]. For melons, F_2:3_ families were obtained from the small-grained seed line ‘P5’ of thin-skinned melons and the large-grained seed line ‘P10’ of thick-skinned melons. Two candidate-related factors, *MELO3C010689* and *MELO3C010690,* were identified [18]. There have also been some studies on seed-size-related traits in cucumbers. The RIL population formed by hybridization of two cultivated cucumber strains (large-seed ‘D06157’ and small-seed ‘D0603’) was used to detect six seed length QTL sites [19]. Fourteen seed-size-related QTL loci were detected on five chromosomes (2, 3, 4, 5, and 6), among which five controlled length QTL loci had PVE values ranging from 7.5% to 15.6%, and four controlled width QTL loci had PVE values of up to 18.8% [20]. At present, the research on seed size of cucurbit crops mainly focuses on genetic analysis and localization of genes controlling seed size. Compared with field crops, such as Arabidopsis and rice, the research on genes related to melon crops seed size is still lacking.

In this study, genetic analysis of seed size in chieh-qua was performed. Combining bulked segregant analysis sequencing (BSA-seq) and fine-mapping, the locus of seed size was finally determined on the 48.8 Kb region of chr10. It contains only one gene, *Bch10G006400*. RNA-seq and variation and expression analysis also suggested that this gene was an important candidate gene for seed size. The results of this study provide a valuable reference for molecular-marker-assisted breeding of chieh-qua seed size and revealing the molecular mechanism of seed size formation.

## 2. Results

### 2.1. Inheritance and Phenotypic Characterization of Seed Size in Chieh-qua

One-way ANOVA results showed that the seed length, width, and quality of the F_1_ generation were significantly different from those of P_2_ (Table 1 and Figure 1a). The three traits (seed length, width, and 30-seed weight) of the F_2_ population were separated and distributed bimodally (Figure 1b–d), indicating that seed size is likely a quantitative trait. Observations of the chieh-qua seeds at different stages revealed that 5–20 d after pollination was the key period of chieh-qua seed development, and after 20 d, the changes in the seeds tended to stabilize (Figure 2).

### 2.2. Preliminary Mapping of the Seed Size Gene

Through whole-genome resequencing of four DNA samples of ‘FJ5’ and ‘J16’, the large-seed pool and the small-seed pool, data quality control revealed a total of 136.06 Gb of clean data, with a Q20 of 99.48% and a Q30 of 96.29%. For all the samples, the average mapping ratio to the reference genome was 99.69%. The average coverage depth was 36.50×. The average genome coverage was 99.18%. The resequencing data were of high quality and could ensure the reliability of the subsequent analysis. Variant identification and SNP identification: A total of 1,610,307 SNPs were identified in the parents, 8090 of which were nonsynonymous mutations. In the offspring pools, 110,399 SNPs were identified, 551 of which were nonsynonymous. By calculating the SNP index, the seed size gene was mapped to 1.31 Mb of 1,548,000–1,679,000 bp on chr10 (Figure 3b). Based on the Euclidean distance (ED), a candidate region with a total length of 13.02 Mb was identified on chr10 (Figure 3a). The two methods intersect to map the seed size gene to a candidate interval of 1.31 Mb in total length on chr10. There were 49 genes in this interval, among which six genes had nonsynonymous mutations.

### 2.3. Fine Mapping of the Seed Size Gene

To verify and narrow the candidate region for seed size identified through BSA-seq analysis, seven polymorphic InDel markers (SS_1 to SS_7) covering the seed size candidate region were developed (Appendix A) and used to genotype the 363 F_2_ individuals of ‘J16 × FJ5’. Based on the genotypes and phenotypes, a molecular linkage map with a genetic distance ranging from 0 cm to 13.80 cm was constructed, and two InDel markers (SS_4 and SS_5) were cosegregated with seed size. Thus, the seed size locus was mapped to a 141 Kb region between markers SS_3 and SS_7 (Figure 4a).

To further narrow the seed size gene, two flanking markers, SS_3 and SS_7, were used to identify 2281 F_2_ plants. As a result, 33 recombinant plants were identified, including six types (Type 1–6). Moreover, four InDel markers were developed (SS_8 to SS_11), and 33 recombinant plants were genotyped. The seed size gene was mapped to a 48.8 Kb region between SS_3 and SS_4 (Figure 4b). Based on the annotation of the referent genome, this fine-mapping region contains only one gene, *Bch10G006400*, which encodes the N-acetyltransferase (NAT) protein. Therefore, it is speculated that *Bch10G006400* is an important candidate gene to determine the seed size of chieh-qua.

### 2.4. RNA-Seq and qRT-PCR Analysis of Candidate Genes

By observing seeds at different development stages, we selected 10 d seeds for RNA sequence analysis. A total of 1974 genes were optimized and 2298 new genes were identified, of which 530 were functionally annotated. No new genes were found in our localized regions. In addition, the RNA-seq data showed a non-identical mutation (G to C), which was consistent with our previous results, and the FPKM value of *Bch10G006400* in ‘J16’ was significantly higher than that of ‘FJ5’ (Figure 5a).

The expression level of *Bch10G006400* in ‘J16’ was significantly higher than that in ‘FJ5’ at 5–20 d, and the expression trend of *Bch10G006400* was consistent with the development trend of litchi seeds. Further analysis showed that *Bch10G006400* was rarely expressed in the roots, stems, and leaves, but its expression level in ‘J16’ flesh was significantly greater than that in ‘FJ5’ flesh (Figure 5b). *Bch10G006400* may also have some effect on flesh. It can be speculated that potential homologous genes controlling seed size and fruit size are highly conserved in structure and function, which may indicate that the QTLs or genes controlling fruit and seed size share the same genetic phenomenon [21].

### 2.5. Sequence Characterization of the Candidate Genes

We subsequently cloned and compared the cDNA sequence of *Bch10G006400* between the two parents. As a result, there are three SNPs within the coding region of *Bch10G006400*, including a nonsynonymous mutation (G to C) at 314 bp, named SNP_314, and two other synonymous mutations at 306 bp and 699 bp (Figure 6a, Appendix A).

At present, *Bch10G006400* has been reported to be related to seed size in wax gourd [22], and the SNP (G to T) at 400 bp, named SNP_400, caused early termination of translation. However, the SNP was not found in the two chieh-qua parents in this study, so we speculated that the mutation of this gene varies among different materials. Twelve chieh-qua germplasms with different genetic backgrounds, including four large seeds and eight small materials, were subjected to whole-genome resequencing (Table 2). As a result, a 3 bp deletion from 551 to 553, named InDel_551, was also found in the small-seeded materials, except for SNP_314 and SNP_400 (Table 2). Subsequently, Sanger sequencing was performed on 50 different germplasm samples from the chieh-qua variety, and the results also confirmed these results (Figure 6b). These three variations formed four haplotypes, namely, G-G-AGA, C-G-AGA, G-T-AGA, and G-G (in the order: SNP_314, SNP_400, and InDel_551). Except for the G-G-AGA haplotype, which had large seeds, the other haplotypes had small seeds. Generally, SNP_314 led to the conversion of proline to arginine (P to R), SNP_400 led to early termination codons, and the deletion at 551–553 led to the deletion of glutamate (E) (Figure 6c). According to the three variations of *Bch10G006400*, two dCAPS markers and one InDel marker were developed to genotype 108 chieh-qua materials. The results showed that the combination of the three markers could distinguish the seed size of the chieh-qua variety with an accuracy rate of 100% (Appendix A).

### 2.6. Transcriptome Analysis of the Seed Size of Chieh-qua

Based on the transcriptome data, we obtained the overall gene expression profiles of ‘J16’ and ‘FJ5’ seeds. There were 2112 differentially expressed genes (DEGs), of which 1009 were upregulated and 1103 were downregulated (Figure 7a). To further explore the metabolic pathways affecting seed size, GO and KEGG enrichment analyses were performed. GO enrichment indicated that the DEGs were involved mainly in biological processes such as carbohydrate metabolic process, cell wall organization, response to oxidative stress, response to oxygen-containing compounds, and response to auxin, among which carbohydrate metabolic process was the most significant. KEGG enrichment indicated that the DEGs were involved mainly in signaling pathways such as plant hormone signal transduction, phenylpropanoid biosynthesis, starch and sucrose metabolism, and pentose and glucuronate interconversions, with plant hormone signal transduction being the most significant (Figure 7b and Appendix A). These findings provide important insight into the molecular mechanism that regulates the seed size of chieh-qua.

## 3. Discussion 

In this study, the length, width, and 30-seed weight of chieh-qua seeds of different generations were measured. The results showed that the seed phenotype of the F_1_ population was similar to that of the large-seed parent P_1_ (Table 1 and Figure 1a), and the seeds of the F_2_ population exhibited a bimodal distribution (Figure 1b–d). These findings are consistent with the results for the seed size of wax gourd [22]. However, some researchers believe that seed size is a quality trait, which may be caused by differences in the materials, planting methods, or statistical methods used.

With the continuous development of second-generation sequencing technology, MutMap [23], QTL-seq [24], QTG-seq [25], and other technologies have been successively reported. Currently, BSA-seq has been applied to characterize the localization of various crops. This method has the advantages of low cost, simple operation, and high resolution for gene identification and mapping. Using cucumber (*Cucumis sativus* L.) F_2_ population as materials, long neck and short neck pools were constructed for high-throughput sequencing and BSA analysis, and a major QTL was identified on cucumber chromosome 7 with a threshold of 0.833 and an interval of 9.98 Mb to 14.61 Mb. The results of BSA mapping and genetic mapping were subsequently combined to map QTLs on chromosome 7 for subsequent fine mapping and functional verification [26]. BSA-seq analysis of the F_2_ population was then used to map the late green locus of watermelon to 7.48 Mb on chromosome 3, after which a large number of CAPS markers were subsequently developed to screen recombinant plants in the F_2_ population. Finally, three candidate genes controlling delayed green leaf color were mapped within a 53.54 Kb region between the SNP130 and SNP135 markers [27]. Analysis of F_2_ and BC_1_ plants by BSA-seq revealed that the skin color gene of wax gourd was located on chr05; a large number of F_2_ plants were screened, after which the number of genes was reduced to 179 Kb, and the gene candidate *Bch05G003950* was identified [28].

In this study, 20 large seeds and 20 small seeds were selected from the F_2_ population to construct a DNA mixing pool, and BSA-seq was performed for the parents and mixing pool. A confidence interval of 1.31 Mb was found on chr10, which was speculated to be the seed size gene locus of chieh-qua (Figure 3). Based on the BSA-seq results, InDel markers were used to genotype 363 F_2_ individuals, and the interval was reduced to 141 Kb (Figure 4a). To further the localization, 33 recombinant plants were selected from 2281 F_2_ plants and divided into six types. Finally, the seed size gene was localized within the 48.8 Kb range and contained only one gene (*Bch10G006400*) (Figure 4b), which encodes a family of N-acetyltransferase proteins. It is speculated that this gene is an important candidate for determining the seed size of chieh-qua, which was consistent with the findings of previous studies of the wax gourd seed size gene [22].

N-acetyltransferase is an amino-acetylated protein whose purpose is to acetylate relatively small molecules. It is one of the most common protein modifications in mammals and is involved in mRNA synthesis regulation [29], DNA damage repair [30], and cell cycle regulation [31]. Several researchers have shown that the *HLS1* gene of *Arabidopsis thaliana* encodes an N-acetyltransferase [32], and that *HLS1* may modify *ARF2* through acetylation to promote its degradation [33]. In addition, *HLS1* regulates the expression of sugar response genes and is related to plant immunity and aging [34]. Therefore, *HLS1* may be a multifunctional molecule in plants. *HLS1* directly interacts with photosensitive pigment interaction factor 4 (*PIF4*) to cause thermal morphogenesis [35,36]. Deletion of *HLS1* and four mutations in three homologous genes in adult plants resulted in abnormal embryogenesis, dwarfing, and flower defects [37,38,39], which indicated that *HLS1* plays a key role in plant development. Recent studies have shown that *HLS1* originates in germ plants, and in addition to the above-mentioned functions, *HLS1* can also delay the flowering of plants. Although *HLS1* has been cloned for more than 20 years, its biochemical mechanism is still unclear [40].

We next cloned the full-length CDS of the parent *Bch10G006400* and found two synonymous mutations and one nonsynonymous mutation (SNP_314), which was consistent with the results of BSA-seq and RNA-seq. Resequencing of 12 materials and Sanger sequencing of 50 materials revealed that, in addition to SNP_314, two other loci (SNP_400 and InDel_551) were present in some small-seeded materials (Table 2, Figure 6). These three loci constitute four haplotypes (G-G-AGA, C-G-AGA, G-T-AGA, and G-G), of which only G-G-AGA is a large seed, and the other three haplotypes are small seeds. It is worth noting that in wax gourd, *Bch10G006400* did not involve the other two loci (SNP_314 and InDel_551), which may be caused by different materials [22]. On the other hand, they verified that most of the materials used were large seed materials, so it was easy to ignore the other two sites where small seed materials existed. Subsequently, two CAPS markers and one InDel marker were developed for genotyping of chieh-qua, which showed 100 percent accuracy (Appendix A). Finally, GO enrichment and KEGG enrichment analyses of the DEGs revealed that they were most significantly related to carbohydrate metabolism and plant hormone signal transduction (Figure 7b, Appendix A). However, the specific mechanism of *Bch10G006400* was unclear, and the underlying molecular mechanism needs to be further studied.

## 4. Materials and Methods

### 4.1. Plant Materials and Phenotyping

Two inbred lines, ‘J16’ (large seed) and ‘FJ5’ (small seed), were used as the maternal parent and the paternal parent, respectively, and an isolated F_2_ population (N = 236) was established for genetic analysis. The large F_2_ population (N = 2281) was used for fine localization of seed size sites. In addition, 108 different species of melon were used to determine the reliability of the results (Appendix A). All materials are provided by Guangzhou Academy of Agricultural Sciences.

The material was planted in the Nansha experimental base of the Guangzhou Academy of Agricultural Sciences (Guangzhou, China, 23.4° N, 113.4° E). Ten full seeds were selected from each plant of 4 generations and their length and width were measured with vernier calipers. The 30-seed weight was then weighed using a 1/10,000th balance.

### 4.2. BSA-Seq Analysis 

DNA from young leaves was extracted using a cetyltrimethylammonium bromide (CTAB)-based protocol [41]. The leaves of 20 large seeds and 20 small seeds were selected from the F_2_ population, the plants in each group were mixed with the same amount of DNA to form a large-seed pool and a small-seed pool, and the final concentration was 40 ng·µL^−1^. The parents and two pools were subsequently sent to Biomarker Technologies (www.biomarker.com.cn, accessed on 7 April 2024) for genome resequencing.

DNA libraries (350 bp) for Illumina sequencing were constructed for each accession according to the manufacturer’s specifications. After DNA library construction, sequencing was performed on an Illumina HiSeq platform with 150 bp read lengths. Raw reads were filtered based on the following criteria: pair-end reads with >10% ‘N’ bases; reads on which more than 50% of the bases have a quality score less than 20 (Phred-like score); sequencing adapter. Finally, high-quality sequences were obtained for subsequent analyses.

Clean reads were mapped onto the reference genome to provide essential information for downstream variant analysis. The software package bwa (mem2 v2.2) was designed for aligning high-throughput sequencing short reads against the reference genome [42]. By positioning clean reads on the reference genome, the sequencing depth, genome coverage, etc., of each sample were estimated, followed by variant identification. The candidate regions related to traits were obtained by calculating the SNP index [24] and Euclidean measure (ED) [43], and the intersection of these two methods was used to determine the candidate regions.

### 4.3. Linkage Mapping Analysis 

To narrow the mapping interval, based on the resequencing data of the parents, primer pairs anchoring the InDel sites within the preliminary candidate region were designed using Primer 5 software. Polymorphic InDel markers were used to genotype the 363 F_2_ individuals, and a molecular marker linkage map was constructed by using QTL IciMapping. The volume of the PCR for the InDel markers was 10 µL, which included 1 µL of the DNA template (50–100 ng·µL^−1^), 1 µL of the forward primer (10 µmol·L^−1^), 1 µL of the reverse primer (10 µmol·L^−1^), 5 µL of a 2 × Go Taq Green Master Mix, and 2 µL of ddH_2_O. The PCR procedure was as follows: 95 °C for 3 min; 34 cycles of 95 °C for 30 s, 55–58 °C for 30 s, and 72 °C for 30 s; 72 °C for 5 min; and 12 °C for the remainder of the study. The PCR products were separated via 6% PAGE.

### 4.4. Fine Mapping Analysis 

To further narrow the scope of mapping, an F_2_ population containing 2281 individuals was constructed, and we used flanking markers to genotype and identify the recombinants. New InDel markers (SS_8 to SS_11) were developed from the flanking marker to determine the genotype of the recombinant plant, and a combination of genotype and phenotype analyses was used to infer the most likely target gene region.

### 4.5. RNA-Seq Analysis 

Tissues from the ‘J16’ and ‘FJ5’ lines were excised and frozen in liquid nitrogen for total RNA extraction. These tissues included the seeds (5 d, 10 d, 15 d, 20 d, 25 d, and 30 d after pollination), roots, stems, leaves, and flesh. The 10d seeds RNA was selected and sent to Biomarker Technologies (www.biomarker.com.cn, accessed on 7 April 2024) for RNA-seq.

High-quality clean data were obtained by removing the reads containing joints and low-quality reads. The location information of the reads in the reference genome was obtained by comparing the clean data with the reference genome of the chieh-qua population using HISAT2 (v2.1.0) [44]. StringTie (v2.0) [45] was used for assembly, and the transcriptome was reconstructed for subsequent analysis. The variable splicing types present in each sample and the corresponding expression levels were obtained via ASprofile (v1.0.4) software [46]. If a region outside the original gene boundary is supported by continuous mapped reads, the untranslated region (UTR) of the gene is extended up and down to correct the gene boundary and optimize the gene. Based on the reference genome sequence of chieh-qua, the mapped reads were spliced and compared with the original genome annotation information to find the original unannotated transcription regions and discover new transcripts and new genes of this species to complement and improve the original genome annotation information. The GO Orthology [47] results of the new genes were analyzed using the integrated database of InterPro. After the amino acid sequence of the new gene was predicted, HMMER (v3.2.1) [48] software was used for comparison with the Pfam [49] database to obtain annotation information for the new gene. Through the maximum flow algorithm, FPKM [50] (fragments per kilobase of transcript per million fragments mapped) was standardized as an indicator of the level of transcript or gene expression. The software DESeq2 (v1.22.1) [51] was used for difference analysis. Finally, the differentially expressed genes were enriched according to GO and KEGG analyses.

### 4.6. qRT-PCR Analysis

Real-time quantitative PCR (qRT-PCR) was used to quantify the developmental and tissue-specific expression of candidate genes. The Eastep Super Total RNA Extraction Kit (Promega, Shanghai, China) and the Eastep RT Master Mix Kit (Promega, Shanghai, China) were used for total RNA isolation and synthesis of the first cDNA strand, respectively, according to the manufacturer’s protocol. The premixed SYBR Green quantitative PCR system was used, and the primer sequences for the internal reference gene and *Bch10G006400* are shown in Appendix A. The values of the three reactions were averaged, and the relative expression was determined using the 2^−ΔΔCt^ method [52]. The experiment was conducted using a CFX384 Real-Time System (Bio-Rad, Hercules, CA, USA).

### 4.7. Candidate Gene Cloning and Sequence Comparison

Total RNA was extracted from the seeds using an Eastep Super Total RNA Extraction Kit (Promega, Shanghai, China). cDNA was obtained by reverse transcription using a cDNA synthesis kit (Promega, Shanghai, China) according to the manufacturer’s instructions. The sequences of the primer pairs used to clone genes encoding genes related to seed size are shown in Appendix A. PCR was performed using Phanta Max Super-Fidelity DNA Polymerase (Vazyme, Nanjing, China) in a 10 µL mixture. Subsequently, all PCR products were purified and cloned and inserted into the pMDTM19-T vector (Takara, Osaka, Japan). Positive colonies of each amplicon (at least three) were randomly selected for sequencing and assembly. All the fragments were sequenced by Beijing Qingdao Biotechnology Co., Ltd. (Beijing, China). DNA and amino acid sequences were compared using DNAMAN 9.0 software.

### 4.8. Development of the dCAPS Maker

Through the variation of sites between CDS sequence, dCAPS (http://helix.wustl.edu/dcaps/dcaps (accessed on 14 December 2023)) was used to find available endonucleases, and primer 5.0 was used to design the primer to be used at the other end (Appendix A). Subsequently, genotyping was performed on 108 chieh-qua materials (28 large variety materials and 80 small variety materials).

## 5. Conclusions

We used BSA-seq and fine mapping to locate the seed size loci to the 48.8 Kb region on chr10, and speculated that this trait was controlled by a gene named *Bch10G006400*. The results were verified by expression, as well as sequence and transcriptome analysis. Then, the KEGG enrichment and GO enrichment of DEG in transcriptome data were analyzed. It was found that the plant hormone signal transduction and carbohydrate metabolic process of DEG were the most significant. These results provide important clues for exploring the molecular mechanism of seed size regulation in chieh-qua.

## Figures and Tables

**Figure 1 ijms-25-04236-f001:**
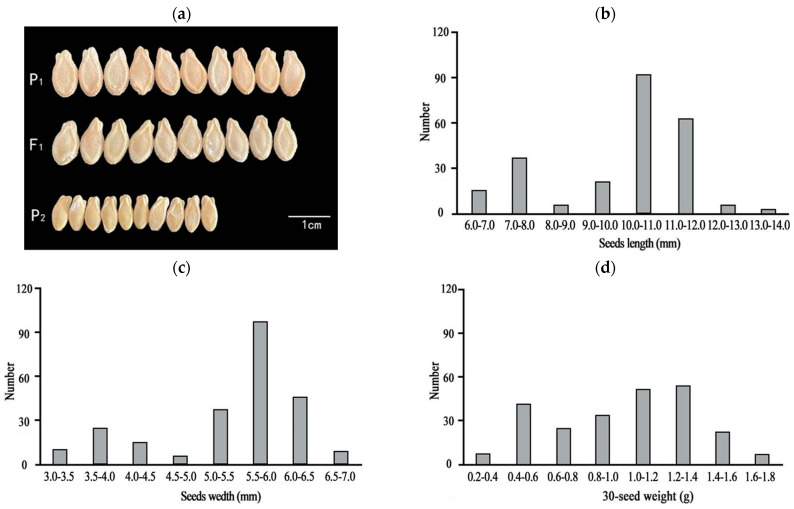
Comparison of parent and F_1_ seeds and the frequency distribution of seed sizes in the F_2_ population. (**a**) Comparison of the sizes of P_1_, P_2_, and F_1_ seeds, with P_1_ and P_2_ representing the parents ‘J16’ and ‘FJ5’, respectively. (**b**–**d**) Distributions of seed length (**b**), width (**c**), and weight (**d**) of 30 seeds in the F_2_ population, respectively.

**Figure 2 ijms-25-04236-f002:**
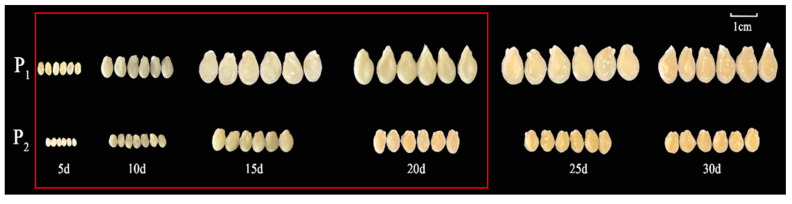
Chieh-qua seeds at different developmental stages. P_1_ and P_2_ indicate ‘J16’ (the female parent) and ‘FJ5’ (the male parent), respectively. The area inside the red box indicates the critical period of seed development.

**Figure 3 ijms-25-04236-f003:**
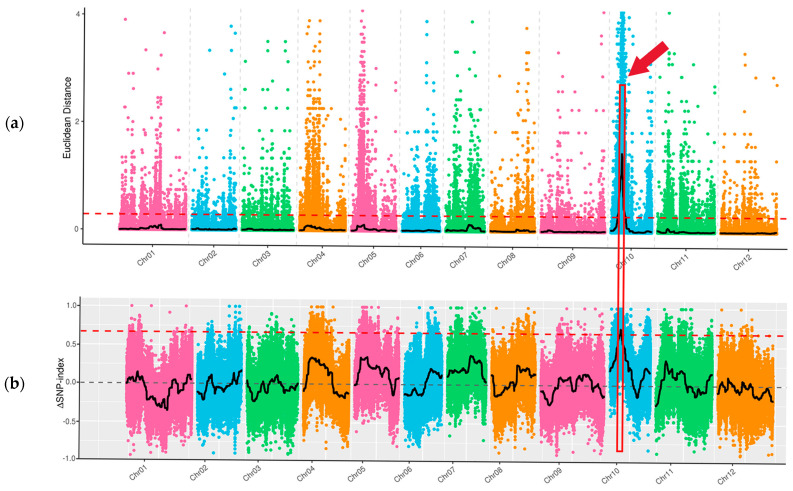
Seed size via BSA-seq mapping of chieh-qua. Note: X-axis is the chromosome ID; colored dots represent SNP index (or ED) values of each SNP site. The black line indicates the fitted SNP index (ED) value. The red dashed line represents the linkage threshold. The region between the red boxes and red arrow indicates the length of the seed size locus. (**a**) Distribution of ED-based linkage values on chromosomes; (**b**) Distribution of SNP indices on chromosomes.

**Figure 4 ijms-25-04236-f004:**
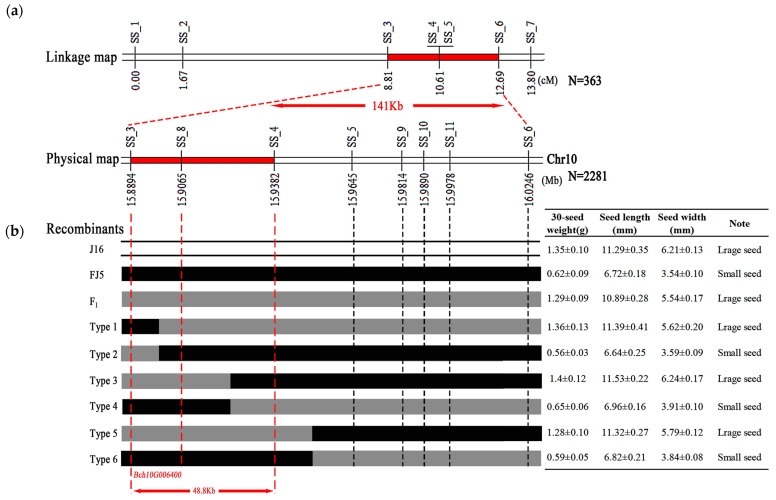
Genetic mapping of the seed size gene in chieh-qua. (**a**) Genetic linkage map of seed size. The red segment denotes the length of the seed size gene locus. The numbers below the bar represent the genetic distance, and SS_1–SS_7 represent the InDel molecular markers. (**b**) Fine-mapping of the seed size gene locus. The region between the red dotted lines indicates the length of the seed size gene locus. The numbers below the bar represent the physical position of chromosome 10. Type 1–Type 6 represent recombinant plant types. The white, black, and gray colors represent the ‘J16’ and ‘FJ5’ genotypes and their F_1_ counterparts, respectively. The table on the right shows the phenotype of each type.

**Figure 5 ijms-25-04236-f005:**
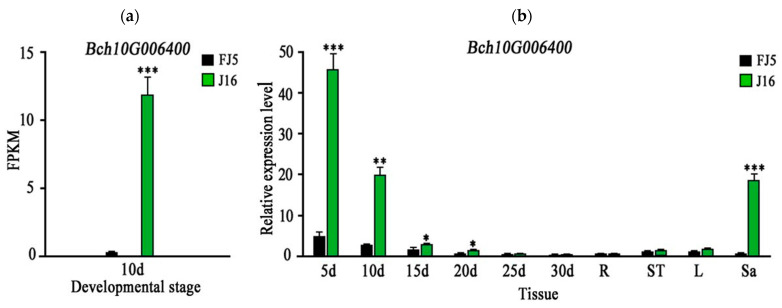
Expression pattern of *Bch10G006400*. The green and black bars indicate the gene expression levels in ‘J16’ and ‘FJ5’, respectively. (**a**) FPKM value of *Bch10G006400* after 10 days. (**b**) 5 d, 10 d, 15 d, 20 d, 25 d, and 30 d represent different stages after pollination. Letters R, St, L, and Sa represent the root, stem, leaf, and sarcocarp, respectively. Error bars indicate the standard errors (SEs). * indicates *p* < 0.05, ** indicates *p* < 0.01, *** indicates *p* < 0.001.

**Figure 6 ijms-25-04236-f006:**
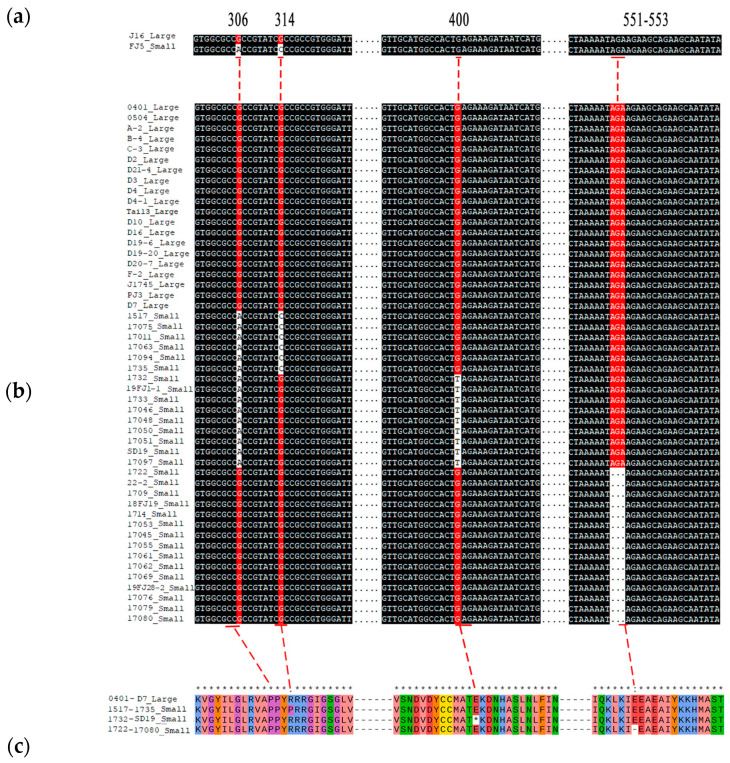
Variations in the nucleotide and amino acid sequences of *Bch10G006400*. (**a**) Comparison of *Bch10G006400* variation sites in parents. (**b**) Comparison of variation sites *Bch10G006400* in 50 chieh-qua materials. (**c**) The four types of amino acid variation sites encoded by *Bch10G006400* were compared, and 0401-D7, 1517-1735, 1732-SD19, and 1722-17080 represent the amino acid sequences encoded by the sequences in (**b**). Large and small represent large and small seeds, respectively. * indicates consistent sequence.

**Figure 7 ijms-25-04236-f007:**
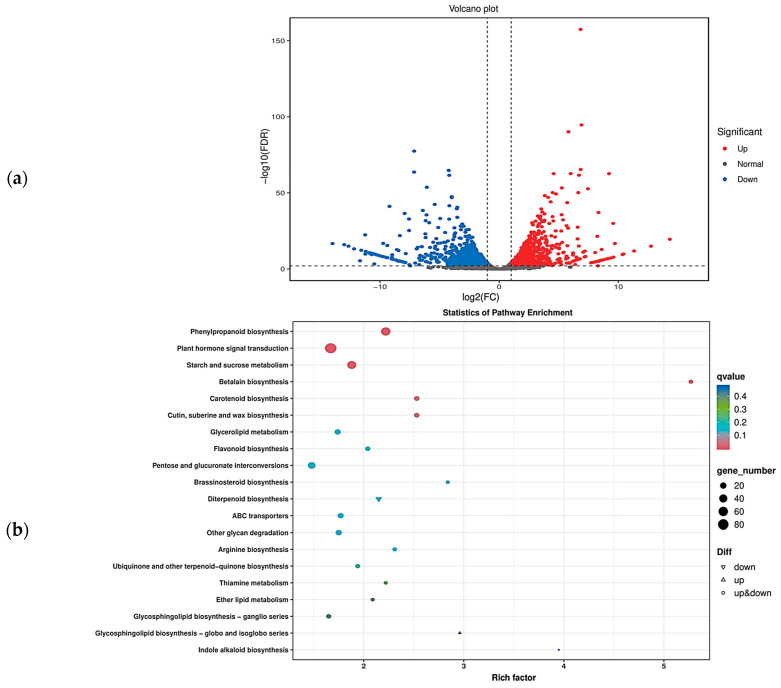
Volcano plot of DEGs and KEGG pathway enrichment. (**a**) Volcano plot of differential expression. In the volcano plot, each dot represents a gene. The blue dots are downregulated genes, the red dots are upregulated genes, and the black dots are genes whose expression did not significantly differ. (**b**) KEGG pathway enrichment of the DEGs in the bubble plot. Each dot represents a KEGG pathway. The color of the dots represents the q value. The smaller the q value is, the more significant or reliable the enrichment. The size of the dots represents the number of DEGs enriched in the pathway. The larger the dot is, the more genes it contains.

**Table 1 ijms-25-04236-t001:** Mean values of seed length, width, and 30-seed weight for three generations of chieh-qua.

Generation	Seed Length (mm)	Seed Width (mm)	30-Seed Weight (g)
P_1_	11.44 ± 0.27 a	5.97 ± 0.14 a	1.34 ± 0.10 a
P_2_	6.50 ± 0.24 b	3.43 ± 0.15 b	0.62 ± 0.07 b
F_1_	11.22 ± 0.81 a	5.83 ± 0.16 a	1.32 ± 0.14 a

Different lowercase letters after the data in the same column indicate significant differences (*p* < 0.05).

**Table 2 ijms-25-04236-t002:** Sequence variation of *Bch10G006400* in 12 chieh-qua cultivars.

Sample ID	SNP_314	SNP_400	InDel_551	Length (mm)	Width (mm)
D2-2-2	G	G	AGA	11.49 ± 0.36	6.46 ± 0.56
D4-1-2	G	G	AGA	12.11 ± 0.57	6.13 ± 0.39
0401	G	G	AGA	11.2 ± 0.28	5.91 ± 0.48
Tai 13	G	G	AGA	11.26 ± 0.58	6.37 ± 0.40
17075	C	G	AGA	6.81 ± 0.44	3.72 ± 0.29
17063	C	G	AGA	4.98 ± 0.49	2.91 ± 0.21
17011	C	G	AGA	6.60 ± 0.84	3.55 ± 0.18
19FJ1-1	G	T	AGA	6.31 ± 0.79	3.44 ± 0.46
17097	G	T	AGA	5.86 ± 0.29	3.11 ± 0.28
19FJ28-2	G	G	-	5.69 ± 0.46	3.19 ± 0.37
17045	G	G	-	5.66 ± 0.41	3.43 ± 0.23
1714	G	G	-	5.68 ± 0.54	3.20 ± 0.37

## Data Availability

The data presented in this study are available in this article and Appendix A.

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
