# Peer review of "Natural Allelic Variations of *Bch10G006400* Controlling Seed Size in Chieh-qua (*Benincasa hispida* Cogn. var. *Chieh-qua* How)"

_ijms, 2024, doi:10.3390/ijms25084236_

Round 1

Reviewer 1 Report

Comments and Suggestions for Authors

The manuscript entitled ,,Natural allelic variations of Bch10G006400 controlling seed size in chiehqua (Benincasa hispida Cogn. var. Chieh-qua How)” is of great scientific value. The obtained results provide important clues for the analysis of the molecular mechanism of seed size regulation in chieh-qua. The documentation is well prepared. The methodology was chosen correctly. The discussion is interestingly written. The standard deviation must be included in Figure 1. English needs to be corrected by a native speaker. In my opinion, after taking into account these corrections, the manuscript can be published.

Comments on the Quality of English Language

English needs to be corrected by a native speaker.

Reviewer 2 Report

Comments and Suggestions for Authors

Bch10G006400 A review of allelic variation in chieh-qua seed size is a pioneering study of the complex genetic mechanisms underlying seed development. Through a careful combination of BSA-seq, genetic mapping and transcriptome evaluation, the researchers have delved into the molecular landscape of chieh-qua seeds, uncovering key insights that promise to revolutionise crop improvement strategies.
One of the biggest first-mover factors in this study is the accuracy with which the researchers identified the seed size locus at the narrow 48.8 Kb site Chr10. By identifying specific SNPs and haplotypes associated with the seed length variant, the study not only sheds light on the genetic shape of chieh-qua seeds, but also paves the way for targeted breeding programmes aimed at increasing seed size and yield.
Furthermore, the differential gene expression assessment and targeted characterisation of Bch10G006400 reveals the importance of this gene in regulating seed size. The approach taken by the researchers, from full genome sequencing to expression profiling, provides a solid basis for understanding the molecular basis of chieh-qua seed size.
Overall, this observation is a good improvement of our knowledge on the control of seed length in flowers. The results of this study not only have implications for chieh-qua breeding but also provide valuable insights that can be extrapolated to other plant species. By unlocking the genetic secrets of seed length regulation, this study raises new requirements for precision breeding and genetic engineering in agriculture.
The manuscript is proposed for adoption and has been expanded and corrected. I wish you all the best for your future work.
